# 2-(2-(Dimethylamino)vinyl)-4*H*-pyran-4-ones as Novel and Convenient Building-Blocks for the Synthesis of Conjugated 4-Pyrone Derivatives

**DOI:** 10.3390/molecules27248996

**Published:** 2022-12-16

**Authors:** Dmitrii L. Obydennov, Diana I. Nigamatova, Alexander S. Shirinkin, Oleg E. Melnikov, Vladislav V. Fedin, Sergey A. Usachev, Alena E. Simbirtseva, Mikhail Y. Kornev, Vyacheslav Y. Sosnovskikh

**Affiliations:** Institute of Natural Sciences and Mathematics, Ural Federal University, 620000 Ekaterinburg, Russia

**Keywords:** 4-pyrone, DMF-DMA, enamination, cycloaddition, merocyanine, 1,6-conjugate addition, solvatochromism, fluorophore

## Abstract

A straightforward approach for the construction of the new class of conjugated pyrans based on enamination of 2-methyl-4-pyrones with DMF-DMA was developed. 2-(2-(Dimethylamino)vinyl)-4-pyrones are highly reactive substrates that undergo 1,6-conjugate addition/elimination or 1,3-dipolar cycloaddition/elimination followed by substitution of the dimethylamino group without ring opening. This strategy includes selective transformations leading to conjugated and isoxazolyl-substituted 4-pyrone structures. The photophysical properties of the prepared 4-pyrones were determined in view of further design of novel merocyanine fluorophores. A solvatochromism was found for enamino-substituted 4-pyrones accompanied by a strong increase in fluorescence intensity in alcohols. The prepared conjugated structures demonstrated valuable photophysical properties, such as a large Stokes shift (up to 204 nm) and a good quantum yield (up to 28%).

## 1. Introduction

4-Pyrones are an important class of compounds that are widely distributed in nature, exhibiting various beneficial biological activities (e.g., phenoxan (Figure 1), which is active against HIV) [1,2] and are also used as multifunctional building blocks for organic synthesis [3,4,5,6,7,8]. On the other hand, the presence of conjugated double bonds in the pyrone structure makes it possible to consider these heterocycles in terms of attractive photophysical properties. Hispidin, as an important styryl pyrone, is responsible for bioluminescence in basidiomycete fungi in the result of oxidation [9,10], and Cyercene A is a photoactive protective agent in marine mollusks [11,12]. It has also been shown that styryl-substituted 4-pyrones find applications as fluorophores [13,14,15,16,17,18,19], exhibiting mechanochromism and solvatochromism based on the aggregation-induced emission enhancement (AIEE) phenomenon [16,17,18] and can act as molecular switches [19].

The introduction of an additional electron-withdrawing substituent into the C-4 position of the pyrone ring by means of the Knoevenagel reaction leads to one of the most popular merocyanines, DCM (Figure 1), and other 4-methylene-4*H*-pyrans, which have found wide application due to their important photophysical properties [20,21,22,23,24].

The major methods for the preparation of 2-vinyl-substituted 4-pyrones are based on the functionalization of the active methyl group based on the aldol condensation with aromatic aldehydes (Figure 1) [13,14,15,16,17,18,19]. In addition, approaches are known that include the decarboxylative rearrangement of dehydroacetic acid derivatives [15] or the Horner-Wadsworth–Emmons reaction, which finds particular use in the synthesis of naturally occurring pyrans [1,12]. To the best of our knowledge, the transformation of 2,6-dimethyl-4-pyrone with DMF-DMA resulted in the monoenamination product with poor yield (5%) [24], whereas β-dimethylaminoacrolein aminal acetals led to 6-bis(4-dimethylaminoalka-l,3-dienyl)-4*H*-pyran-4-ones [13].

It is important to note that the methods for functionalizing 4-pyrones using nucleophilic reagents to create strong push-pull systems are scarcely studied. Typically, such reactions proceeded via ring opening transformation to form new cyclic systems [25,26,27]. The introduction of the enamino moiety [28,29,30,31,32,33,34] into 4-pyrone molecules leads to new highly reactive substrates, which can be used for the creation of valuable dyes via modification of 4-pyrone moiety or enamino group. Despite their attractiveness, 4-pyrone-based fluorophores are severely limited and have been described in just a few papers [13,14,15,16,17,18,19] due to the modest photophysical properties of the heterocycles and being overshadowed by derived dyes, 4-methylene-4*H*-pyrans.

In this paper, we describe a general strategy to 4-pyrone-bearing merocyanines based on an enamination with DMF-DMA and subsequent transformation of the dimethylaminovinyl group via a nucleophilic 1,6-addition or cycloaddition reaction. This approach opens straightforward access to a wide range of new promising pyran fluorophores.

## 2. Results and Discussion

### 2.1. Synthesis of 2-Enamino-substituted 4-Pyrones and Their Chemical Properties

The functionalization of the pyrone ring was carried out via an enamination reaction at the active methyl group using DMF-DMA as a reagent and a solvent (Figure 2) [35]. *N*-Methylimidazole (NMI) was selected as a convenient base for the promotion of the transformation [36]. Enamination of 2-(*tert*-butyl)-6-methyl-4*H*-pyran-4-one (**1a**) with DMF-DMA (3 equiv.) and NMI (3 equiv.) at 100 °C in an autoclave afforded enamino-substituted 4-pyrone **2a** in only 15% yield (Table 1, entry 1). We decided to increase the reaction temperature to 120 °C and study the influence of the base amount on both the reaction outcome and time (TLC monitoring). When one equivalent of NMI was used, the reaction was completed in 15 h and the product was prepared in 40% yield (entry 2). We found that a further decrease in the amount of NMI (0.25–0.5 equivalent) allowed the improvement of the enamination reaction outcome until 67–72%, but it required longer heating (20–25 h) (TLC monitoring) (entries 3,4). The best yield (72%) of pyrone **2a** was achieved using 0.25 equivalents of *N*-methylimidazole though it took 25 h (entry 4). The isolation of the pyrone included simple recrystallization from *n*-heptane. Interestingly, the reaction also occurred without promotion of the base and gave the product in a lower yield (57%) under heating at 120 °C for 25 h (entry 5). Increasing temperature to 130 °C led to pyrone **2a** in 54% yield (entry 6). The enamination with the use of pyridine as a solvent and DMF-DMA (1.2 equiv.) at 100 °C or 120 °C did not produce the desired product.

The enamination reaction conditions were extended for various 2-methyl-4-pyrones (Figure 3, Table 2), but this transformation turned out to be very sensitive to the nature of substituents at the pyrone ring. The enamination of 2-methyl-6-phenyl-4-pyrone (**1b**) with DMF-DMA proceeded for 12 h under the optimized conditions; as a result, pyrone **2b** was obtained in 53% yield. However, the reaction of 2-methyl-6-trifluoromethyl-4-pyrone (**1c**) was completed in 5 h at 120 °C, leading to the desired product in only low yield (12%). This result can be explained by side processes due to the presence of the trifluoromethyl group and high CH-acidity). Lowering the temperature to 100 °C made it possible to increase the yield up to 43%.

It was found that the enamination of ethyl 6-methylcomanate (**1d’**) with DMF-DMA and an excess of NMI (3 equiv.) at 120 °C was accompanied by the transesterification reaction to produce product **2d** in 27% yield (Figure 3 Table 2). It is interesting to note that direct enamination of methyl 6-methylcomanate (**1d**) at 100 °C in the presence of NMI (0.25 equiv.) led to the desired product **2d** in only 8% yield.

We tried to extend the enamination on 2-methyl-6-styryl-4-pyrones **1e–g** for the synthesis of unsymmetrical 2,6-divinyl-4-pyrones **2e–g** (Table 2). The reaction of 2-methyl-6-styryl-4-pyrone (**1e**) at 120 °C with a threefold excess of DMF-DMA and different amounts of *N*-methylimidazole did not lead to the desired product. Lowering the temperature to 100 °C made it possible to obtain product **2e** in 22% yield for 4 h (TLC monitoring). For starting 2-methyl-6-styryl-4-pyrones **1f,g**, the use of the optimized conditions did not give the desired products because of low solubility of the starting materials. The enamination of pyrones **1f,g** with threefold excess of *N*-methylimidazole and heating at 120 °C for 3 h led to complete conversion of the starting 4-pyrone, and products **2f,g** were isolated by the treatment with diethyl ether in 72 and 51% yields, respectively. Such a difference in the behavior of styrylpyrones may be connected with the presence of electron-donating substituents, which deactivated the double bond of the styryl fragment and reduced the possibility of side reactions.

The reaction of 2,6-dimethyl-4-pyrone **3a** with three equivalent DMF-DMA and 0.5 equiv. of NMI was carried out at 120 °C for 15 h, resulting in a mixture of products of enamination **4a** and **5a** (Figure 4). Recrystallization from *n*-heptane easily allowed the separation of pyrones **4a** and **5a** and isolation them in pure form in 17% and 23% yields, respectively. All our attempts to carry out more selective monoenamination by lowering the temperature, variation of reagent amounts, and increasing the reaction time were unsuccessful and accompanied by incomplete conversion and the formation of the bis(enamino) derivative, which did not allow the preparation of product **4a** in pure form directly.

The use of an excess of DMF-DMA (5 equiv.) and heating at 130 °C for 15 h made it possible to increase the yield of bisenamine **5a** to 51%. At the same time, the formation of monoenamino derivative **4a** was not observed. Carrying out the reaction without using NMI or using one equivalent of NMI resulted in product **5a** in 41 and 45% yields, respectively. The transformation of 3-bromo-2,6-dimethyl-4-pyrone (**3b**) led to a double enamination product **5b** in 53% yield. This result can probably be explained by a higher CH-acidity of the methyl groups. We also managed to carry out monoenamination selectively at the methyl group located near the electron-withdrawing bromine atom, and pyrone **4b** was prepared in high yield (80%) (Figure 4). The structure of the product was assigned on the basis of the chemical shift of the methyl group in comparison with the starting 2,6-dimethyl-3-bromo-4-pyrone (**3b**) and product **4b**.

The ^1^H NMR spectra of the obtained dimethylamino-substituted pyrones **2,4,5** demonstrate a characteristic set of two doublets of the enamino group with ^3^*J* coupling of 12.6–13.3 Hz, which indicates the *E*-configuration of the double bond and its partially double order due to the strong push-pull nature [31,36].

To study the chemical properties of monoenamino-substituted 4-pyrones **2** with various nucleophilic reagents, compound **2g** was used as an example to obtain conjugated structures (Figure 5). We found that heating in AcOH turned out to be convenient conditions for carrying out the reactions. The transformation of pyrone **2g** with aniline or diphenylamine at 90 °C led to the substitution of the dimethylamino group and the formation of products **6a,b** in 82–86% yield. The reaction of substance **2g** with *p*-phenylenediamine as a binucleophile gave product **6c** as the result of an attack on both amino groups. It was found that pyrone **2g** reacted with benzylamine under reflux in acetonitrile to form product **6d** in 76% yield. The transformation of bis(enamino)pyrone **5a** with aniline was found to proceed at room temperature, leading to product **6e** in 55% yield. Thus, it has been shown that the side chain of γ-pyrone can easily be functionalized with aliphatic and aromatic amines.

Enamino-substituted pyrones **2g** and **5a** were able to react with 2-methylindole as a C-nucleophile to form indolyl-substituted 4-pyrones **7a,b** in 52–62% yields under reflux in AcOH for 7–10 h (Figure 5). The transformation of bisenamine **5a** with 2-methylindole included the substitution at two enamino fragments and led to bis(indolylvinyl)-4-pyrone **7b** in 52% yield.

Next, we investigated the cycloaddition of enamino-substituted 4-pyrones with 1,3-dipoles (Figure 6). It was observed that organic azides and diphenylnitrilimine did not give the desired products, which is probably due to the electron-withdrawing properties of the pyrone ring. The reaction with benzonitrile oxide, which was generated in situ from *N*-hydroxybenzimidoyl chloride [25], in dioxane without the use of a base led to the formation of isoxazolyl-substituted 4-pyrones in 39–80% yields. Although compounds **2f,g** bear two double bonds of different nature, the transformation proceeded chemoselectively at the enamino fragment. In the case of bis(enamino) derivative **5a**, the cycloaddition occurred at both enamino moieties to give 2,6-bis(isoxazolyl)-4-pyrone **9** in 39% yield.

### 2.2. Photophysical Properties of Products

For the series of enamino-substituted 4-pyrones, the photophysical properties were studied to assess the prospects for their use as fluorophores. We started with the study of the influence of the nature of the solvent on the absorption and emission spectra of (*E*)-2-(2-(dimethylamino)vinyl)-6-methyl-4*H*-pyran-4-one (**4a**) and 2,6-bis((*E*)-2-(dimethylamino)vinyl)-4*H*-pyran-4-one (**5a**).

For the monoenamino-substituted compound **4a**, the absorption spectrum includes one-band at 334–363 nm with an extinction coefficient of 29,200–35,900 M^−1^cm^−1^ (Figure 2, Table 3). In aprotic solvents, the absorption maximum is observed at 334–350 nm. For alcohol solutions of pyrone **4a**, the absorption maximum shifts slightly to the long-wavelength region and appears at 356 (*i*-PrOH), 361 (EtOH), 363 nm (MeOH) in accordance with the solvent polarity. The emission spectrum demonstrates a single maximum and depends strongly on the nature of the solvent. In alcohols as protic solvents, the fluorescence intensity increases many times over in comparison with aprotic polar solvents, such as DMSO. The highest values of quantum yields are achieved in MeOH (3.6%) and EtOH (1.4%), where the substance exhibits blue fluorescence. The largest Stokes shifts (67–71 nm) are also observed in MeOH and EtOH, while it is equal to 26–59 nm in the other solvents. The peculiarity of fluorescence in protic solvents can be related to the specific solvation of carbonyl oxygen in the excited state due to intramolecular charge transfer (ICT). The solvatochromism of 4-pyrones was previously unknown and distinguishes the studied conjugated 4-pyrones from 4-methylene-4*H*-pyrans, popular merocyanine dyes whose fluorescence is related to the solvent polarity and is most pronounced in DMSO.

In the case of bis(enamino) derivative **5a**, two maxima are observed in the absorption spectra. The most intense and structured band is in the short wavelength region at 300–304 nm (ε = 41,400–54,400 M^−1^ cm^−1^), and at 378–408 nm there is a second maximum with an extinction coefficient of 17,400–22,300 M^−1^ cm ^−1^(Figure 3, Table 4). The nature of the solvent most strongly affects the second maximum, which can be associated with intramolecular electron transfer. The strongest redshift of the second band is observed in alcohols (395–408 nm) compared to aprotic solvents (378–381 nm). As in the case of monoenamino derivative **4a**, the fluorescence spectra turned out to be highly sensitive to the nature of the solvent and has one emission maximum located in the range of 455–490 nm. The fluorescence intensity in polar aprotic solvents is observed to be low (QY = 2.3–4.1%). The substance exhibits green fluorescence, and the highest quantum yield is found in methanol (QY = 28%), ethanol (QY = 21%), and isopropanol (QY = 11%). Also, in these solvents, the largest Stokes shifts are observed, which are equal to 83 nm, 82 nm, and 80 nm, respectively. 

For the design of new fluorophores, we studied a number of functionalized enamino-substituted 4-pyrones (Figure 4, Table 5). The introduction of the *tert*-butyl group, compared to the methyl group, has practically no effect on the photophysical properties. The absorption and emission spectra of (*tert*-butyl)-6-(2-(dimethylamino)vinyl)-4*H*-pyran-4-one (**2a**) are very similar for monoenamine derivative **4a**. The introduction of the phenyl group complicates the absorption spectrum, as a result, several maxima are observed. In this case, the major absorption maximum appears at 380 nm with an extinction coefficient of 19,700 M^−1^cm^−1^. An important feature of fluorescence is the large Stokes shift, which amounts to 147 nm (QY = 3.4%). For dimethylenamino derivative **2f** bearing the *p*-dimethylaminostyryl moiety, the absorption spectra show several maxima with approximately the same value of the extinction coefficient. The fluorescence spectrum exhibits one-band emission at 572 nm (λ_ex_ = 368 nm), which is characterized by higher quantum yield (18%) and a significant Stokes shift (204 nm). Similarly, the *p*-MeO-styryl-substituted compound **2g** has several bands with low intensity in absorption spectra and very weak fluorescence emission.

Introduction of the diphenylamino substituent allows the improvement of photophysical properties. Thus, for *p*-MeO-styryl derivative **6b**, two intense maxima are found in the absorption spectrum at 404 nm and 350 nm. This substance shows one maximum in the emission spectrum at 546 nm (λ_ex_ = 368 nm) and a quantum yield of 3.4%. Next, the bis(indolyl) derivative **7b** was investigated as a symmetrical compound with an extended conjugation system. Its absorption spectrum (MeOH) contains a major maximum at 433 nm, a plateau in the range of 375–348 nm, and a minor maximum at 286 nm. The emission spectrum contains one band at 571 nm, while the quantum yield reaches 15%.

### 2.3. Theoretical Calculations of the Absorption and Emission

We performed a DFT/TD-DFT quantum chemical calculations of absorption/emission maxima for representative compounds **4a** and **5a** in vacuo and in solvated phase (DMSO, EtOH, and MeOH) using the conductor-like polar continuum model (C-PCM). In the series of the ground state (GS), the first singlet excited state (S_1_) optimizations were made and the energies of the first seven Franck-Condon singlet states were computed. All calculations were carried out at the (TD-)DFT (CAM-)B3LYP/6-31++G** level of theory for the most stable *s-trans* conformations [13]. Results of this calculations are provided in Table 6; the optimized geometries of GS and S_1_ are listed in Appendix A.

The optimizations of the ground state (GS) geometry revealed that both molecules have a planar structure of D-π-A conjugation chains. Calculated Stokes shift values in a solvated phase were 27–30 nm for 2-(2-(dimethylamino)vinyl)-6-methyl-4-pyrone (**4a**) and 92–103 nm for 2,6-bis(2-(dimethylamino)vinyl)-4-pyrone (**5a**), which are in agreement with the experimental Stokes shifts. Also, in all solvents the dipole moment values increased under excitation to S_1_. On the base of Franck-Condon (FC) excitations and S_1_ optimized geometries energies and their oscillator strengths, we plotted theoretical absorption and emission spectra (See Appendix A, absorption/emission maxima are also provided in Appendix A). Because the DFT usually overestimates long-wavelength polymethine transitions, all calculated maxima are notably blue-shifted compared to the experimental ones [38]. According to TD-DFT calculations of S_1_^FC^–S_7_^FC^ states in all solvents, the most intense absorption corresponds to π → π* transition and S_0_ → S_1_ from the highest occupied molecular orbital (HOMO) to the lowest unoccupied molecular orbital (LUMO) for compound **4a**, whereas S_0_ → S_2_ (HOMO–1 → LUMO) transition prevails for compound **5a** (Table 6). In the case of compound **5a**, the S_0_ → S_1_ transition is less intense (f_GS_ = 0.527 for MeOH) and redshifted (λ_abs_ = 376 nm for MeOH), which agrees well with the experimental absorption spectrum demonstrating double bands.

To explain the strong influence of alcohols on fluorescence, the charges on the carbonyl oxygen of pyrones **4a** and **5a** in the ground and excited states were compared. An increase in the electron density on oxygen can cause its specific solvation due to the formation of hydrogen bonds, leading to the improvement of the fluorescence properties. The charge of pyrone **5a** bearing two electron-donating substituents is in all cases higher than that of pyrone **4a** in the corresponding solvent, which can be explained by the stronger push-pull nature of the former. In addition, the maximum negative charge was found in EtOH and MeOH, both in the ground (−0.731 and −0.746 for **4a**; −0.758 and −0.776 for **5a**) and excited (−0.823 and −0.839 for **4a**; −0.905 and −0.925 for **5a**) states (Table 7). For all excited states, an increase in the negative charge on carbonyl oxygen is observed (except for **4a** in vacuum), which is in good agreement with the change in dipole moments. The strongest charge changes during the GS→S_1_ transition were found in MeOH (0.093 for **4a**, 0.149 for **5a**). This result is related to the ICT effect in this solvent, which is more pronounced for pyrone **5a** and determines the stabilization of the excited states via the hydrogen bonding interaction between the carbonyl group and alcohol molecules [39]. Besides, the ICT phenomena for the pyrones were confirmed with the use of electron density difference (EDD) maps (See Appendix A).

The analysis of the frontier MOs for the ground states in the solvents (DMSO, MeOH, EtOH) and the gas phase showed that both HOMO and LUMO frontier orbitals are localized chiefly on the polymethine chain atoms, which is known to be typical for merocyanine dyes [22] (Table 8, See Appendix A). Opposite to mono(enamino) derivative **4a**, bis(enamino) derivative **5a** exhibited a complete absence of the electron density at the carbonyl moiety of the conjugation chain for the HOMO, whereas the LUMO localization involves the whole π-conjugation, indicating the ICT effect under excitation. This feature can be connected with a large Stokes shift of bis(enamino) derivative **5a** and the red-shifted emission compared to **4a**.

Thus, a convenient method of 4-pyrone functionalization is developed via enamination of 2-methyl-4-pyrones with DMF-DMA. Enamino-substituted 4-pyrones were able to react with nucleophiles and 1,3-dipoles with the substitution of the dimethylamino group and the formation of conjugated push-pull and isoxazolyl-substituted 4-pyrones, respectively. The transformations occured with high chemoselectivity without 4-pyrone ring opening and provided a convenient platform for the synthesis and design of 4-pyrone-based fluorophores. For the first time, the solvatochromism of 4-pyrones in protic solvents due to specific solvation was discovered, leading to a strong increase in the fluorescence intensity compared to aprotic solvents. 4-Pyrones bearing two enamino fragments show a higher quantum yield and significant Stokes shift, which can be explained by their stronger push-pull character. The prepared pyrone merocyanines are of interest due to attractive photophysical properties and easy modification of the conjugated chain, which can contribute to the development of the synthesis of new organic fluorophores.

## 3. Materials and Methods 

NMR spectra were recorded on Bruker DRX-400 (Bruker BioSpin GmbH, Ettlingen, Germany, work frequencies: ^1^H, 400 MHz; ^13^C, 101 MHz; ^19^F, 376 Hz), Bruker Avance-400 (Bruker BioSpin GmbH, Rheinstetten, Germany, work frequencies: ^1^H, 400 MHz; ^13^C, 101 MHz), Bruker Avance III-500 (Bruker BioSpin GmbH, Rheinstetten, Germany, work frequencies: ^1^H, 500 MHz; ^13^C, 126 MHz), and Bruker Avance NEO (Bruker BioSpin GmbH, work frequencies: ^1^H, 600 MHz; ^13^C, 151 MHz) spectrometers in DMSO-*d*_6_ or CDCl_3_. The chemical shifts (*δ*) are reported in ppm relative to the internal standard TMS (^1^H NMR), C_6_F_6_ (^19^F NMR), and residual signals of the solvents (^13^C NMR). IR spectra were recorded on a Shimadzu IRSpirit-T (Shimadzu Corp., Kyoto, Japan) spectrometer using an attenuated total reflectance (ATR) unit (FTIR mode, diamond prism); the absorbance maxima (*ν*) are reported in cm^−1^. Electron absorption spectra were obtained with a Shimadzu UV-1900 (Shimadzu Corp.) spectrophotometer; fluorescence spectra were obtained with a Shimadzu RF-6000 (Shimadzu Corp.) fluorescence spectrophotometer.

Mass spectra (ESI-MS) were measured with a Waters Xevo QTof instrument (Waters Corp., Milford, MA, USA). Elemental analyses were performed on an automatic analyzer PerkinElmer PE 2400 (Perkin Elmer Instruments, Waltham, MA, USA). Melting points were determined using a Stuart SMP40 melting point apparatus (Bibby Scientific Ltd., Stone, Staffordshire, UK). Column chromatography was performed on silica gel (Merck 60, 70–230 mesh). All solvents used were dried and distilled by standard procedures. 2-Methyl-6-styryl-4-pyrones [15], esters of 6-methyl-4-pyrone-2-carboxylic acid [40,41], 2-*tert*-butyl-6-methyl-4-pyrone [42], and 2-methyl-6-phenyl-4-pyrone [42] were prepared according to the literature procedure.

### 3.1. Quantum Mechanical Calculations

The ground state molecular geometry of the compounds under investigation was fully optimized at density functional theory (DFT) level, both in vacuo and in the solvated phase (DMSO, EtOH, MeOH). For all geometry optimizations, the B3LYP hybrid functional [43] coupled with the 6-31G(d,p)++ basis set was chosen. Solvent effects were taken into account via the implicit conductor-like polarizable continuum model (C-PCM). For the evaluation of energetics, Solvation Model Density (SMD) parametrization was employed [44]. The vibrational frequencies and thermochemicals were computed in harmonic approximation at T = 298.15 K and p = 1 atm, and no imaginary frequencies were found.

The UV-vis absorption spectra for the equilibrium geometries were calculated at time-dependent density functional theory (TD-DFT) level, accounting for S_0_→ S_n_ (n = 1 to 7). The nature of the vertical excited electronic state was analyzed both in vacuo and in the solvated phase.

The first singlet excited state (S_1_) geometry was optimized using analytical gradients and the first transitions S_1_→S_0_ of the emission. Properties of the excited states were calculated using the long-range corrected functional CAM-B3LYP [45,46] coupled with the 6-31G(d,p)++ basis set. The non-equilibrium solvation regime was set for vertical excited states calculations in the solvent phase, whereas the equilibrium solvation was used for adiabatic ones. All calculated UV-vis spectra were plotted as Gaussian curves with wavelengths of absorption/emission maxima as an expected value and σ = 0.4 eV.

The integration grid for the calculations was set to 96 radial shells and 302 angular points.

The RMS gradient convergence tolerance was set to 10^−7^ Hartree/Bohr for GS optimizations and to 10^−5^ Hartree/Bohr for S_1_ optimizations. The density matrix convergence threshold for the self-consistent field was set to 10^−5^ a.u. for all DFT and to 10^−6^ a.u. for all TD-DFT optimizations.

All calculations were performed using the US GAMESS (ver. 30 September 2021, R2 Patch 1) software package for Linux x64 [47]. Frontier MOs were plotted with MacMolPlt software (ver. 7.7) [48]. Electron density difference maps were calculated with the use of Multiwfn v3.8 [49].

### 3.2. Synthesis of Compounds **2**

Corresponding 4-pyrone **1** (1.2 mmol) was heated with DMF-DMA (429.0 mg, 3.6 mmol), *N*-methylimidazole (0.3 mmol for **2a–c,e**; 4.8 mmol for **2d**; 3.6 mmol for **2f,g**) in an autoclave at 100 °C (for **2c**,**e**) or 120 °C (for **2a,b,d,f,g**) for the needed time. For pyrones **2a–c**, the reaction mixture was treated by boiling *n*-heptane (40 mL). The solvent was decanted and evaporated to 2 mL, and the solid was filtered. For products **2d,f,g**, the reaction mixture was treated with Et_2_O and the solid that formed was filtered. Pyrone **2e** was isolated by flash-chromatography with the use of CHCl_3_ as an eluent.

*(E)-2-(Tert-butyl)-6-(2-(dimethylamino)vinyl)-4H-pyran-4-one* (**2a**). The reaction was carried out for 15 h. Yield 191.2 mg (72%), yellow powder, mp 128–130 °C. IR (ATR) ν 2962, 2907, 2870, 1639, 1564, 1386, 1364, 1104. ^1^H NMR (500 MHz, DMSO-*d_6_*) *δ* 1.23 (9H, s, *t*-Bu), 2.90 (6H, br.s, NMe_2_), 4.82 (1H, d, *J* = 13.2 Hz, =CH(α)), 5.62 (1H, d, *J* = 2.1 Hz, CH), 5.77 (1H, d, *J* = 2.1 Hz, CH), 7.26 (1H, d, *J* = 13.2 Hz, =CH (β)). ^13^C NMR (126 MHz, CDCl_3_) *δ* 28.1, 35.8, 40.8 (br.s. NMe_2_), 88.0, 104.8, 108.9, 144.8, 165.8, 172.8, 180.5. Anal. Calculated for C_13_H_19_NO_2_: C 70.56; H 8.65; N 6.33. Found: C 70.58; H 8.76; N 6.33.

*(E)-2-(2-(Dimethylamino)vinyl)-6-phenyl-4H-pyran-4-one* (**2b**). The reaction was carried out for 12 h. Yield 153.5 mg (53%), yellow powder, mp 200 °C (destr.). IR (ATR) ν 3056, 2910, 1627, 1539, 1381, 1349, 1101. ^1^H NMR (400 MHz, DMSO-*d*_6_) *δ* 2.95 (6H, s, NMe_2_), 4.92 (1H, d, *J* = 13.3, =CH(α)), 5.78 (1H, d, *J* = 2.2 Hz, H-3), 6.61 (1H, d, *J* = 2.2 Hz, H-5), 7.47 (1H, d, *J* = 13.3 Hz, =CH(β)), 7.52 (3H, m, Ph), 7.94 (2H, m, H-2, H-6 Ph). ^13^C NMR (101 MHz, DMSO-*d*_6_) *δ* 86.9, 104.3, 110.1, 126.1, 129.4, 131.1, 132.0, 146.7, 160.4, 166.3, 178.3 (NMe_2_ was not observed). HRMS (ESI) m/z [M + H]^+^. Calculated for C_15_H_16_NO_2_: 242.1189. Found: 242.1103.

*(E)-2-(2-(Dimethylamino)vinyl)-6-(trifluoromethyl)-4H-pyran-4-one* (**2c**). The reaction was carried out for 5 h. Yield 120.3 mg (43%), grey powder, mp 71–72 °C. IR (ATR) ν 3071, 2909, 1669, 1557, 1341, 1267, 1078, 959. ^1^H NMR (400 MHz, CDCl_3_) *δ* 2.99 (6H, s, NMe_2_), 4.75 (1H, d, *J* = 12.6 Hz, =CH(α)), 5.84 (1H, s), 6.84 (1H, s), 7.18 (1H, d, *J* = 12.6 Hz, =CH(β)). ^19^F NMR (376 MHz, CDCl_3_) *δ* 90.2 (s, CF_3_). ^13^C NMR (126 MHz, DMSO-*d_6_*) *δ* 36.8 (br.s), 44.0 (br.s), 84.9, 103.5, 113.6 (q, *J* = 2.2 Hz, C-3), 118.7 (q, *J* = 273.2 Hz, CF_3_), 147.6, 148.6 (q, *J* = 38.1 Hz, C-2), 166.9, 175.3. HRMS (ESI) m/z [M + H]^+^. Calculated for C_10_H_11_F_3_NO_2_: 234.0742. Found: 234.0735.

*Methyl (E)-6-(2-(dimethylamino)vinyl)-4-oxo-4H-pyran-2-carboxylate* (**2d**). The reaction was carried out for 6 h. Yield 72.3 mg (27%), yellow powder, mp 130–132 °C. IR (ATR) ν 3064, 2909, 1745, 1632, 1550, 1351, 1098, 959. ^1^H NMR (500 MHz, CDCl_3_) *δ* 2.97 (6H, s, NMe_2_), 4.76 (1H, d, *J* = 13.0 Hz, =CH(α)), 5.89 (1H, d, *J* = 2.3 Hz, H-5), 6.88 (1H, d, *J* = 2.3 Hz, H-3), 7.32 (1H, d, *J* = 13.0 Hz, =CH(β)). ^13^C NMR (126 MHz, CDCl_3_) *δ* 40.1 (br.s, NMe_2_), 53.1, 86.9, 106.9, 118.7, 146.9, 150.0, 161.1, 166.8, 178.7. Anal. Calculated for C_11_H_13_NO_4_: C 59.41; H 5.72; N 5.91. Found: C 59.19; H 5.87; N 6.27.

*2-((E)-2-(Dimethylamino)vinyl)-6-((E)-styryl)-4H-pyran-4-one* (**2e**). The reaction was carried out for 4 h. Yield 70.6 mg (22%), yellow powder, mp 75–77 °C. IR (ATR) ν 3055, 1642, 1610, 1524, 1397, 1157, 1103, 749. ^1^H NMR (500 MHz, CDCl_3_) *δ* 2.98 (6H, s, NMe_2_), 4.79 (1H, d, *J* = 13.1 Hz, -CH=C-N), 5.84 (1H, d, *J* = 1.4 Hz, CH), 6.12 (1H, d, *J* = 1.4 Hz, CH), 6.66 (1H, d, *J* = 16.1 Hz, -CH=C-Ar), 7.24 (1H, d, *J* = 13.1 Hz, =CH-N), 7.27 (1H, d, *J* = 16.1 Hz, =CH-Ar), 7.39 (2H, t, *J* = 7.3 Hz, H-3, H-5 Ph), 7.51 (2H, d, *J* = 7.3 Hz, H-2, H-6 Ph). ^13^C NMR (126 MHz, CDCl_3_) *δ* 40.8, 87.9, 105.6, 113.4, 120.6, 127.3, 128.9, 129.3, 134.1, 135.3, 145.5, 159.5, 165.7, 179.9. HRMS (ESI) m/z [M + H]^+^. Calculated for C_17_H_18_NO_2_: 268.1338. Found: 268.1342.

*2-((E)-4-(Dimethylamino)styryl)-6-((E)-2-(dimethylamino)vinyl)-4H-pyran-4-one* (**2f**). The reaction was carried out for 3 h. Yield 268.2 mg (72%), brown crystals, mp 205–206 °C. IR (ATR) ν 2802, 1596, 1520, 1386, 1357, 1154, 1100. ^1^H NMR (400 MHz, DMSO-*d*_6_) *δ* 2.97 (12H, s, 2NMe_2_), 4.87 (1H, d, *J* = 13.2 Hz, -CH=C-N), 5.65 (1H, d, *J* = 2.1 Hz, CH), 5.93 (1H, d, *J* = 2.1 Hz, CH), 6.69 (1H, d, *J* = 16.2 Hz, -CH=C–Ar), 6.73 (2H, d, *J* = 8.8 Hz, H-3, H-5 Ar), 7.36 (1H, d, *J* = 16.2 Hz, =CH–Ar), 7.48 (1H, d, *J* = 13.2 Hz, =CH–N), 7.53 (2H, d, *J* = 8.8 Hz, H-2, H-6 Ar). ^13^C NMR (101 MHz, DMSO-*d*_6_) *δ* 39.7, 86.6, 103.8, 111.1, 111.9, 115.0, 122.9, 128.9, 134.2, 146.1, 150.9, 160.1, 165.2, 177.9 (NMe_2_ was not observed). Anal. Calculated for C_19_H_22_N_2_O_2_: C 73.52; H 7.14; N 9.03. Found: C 73.47; H 7.27; N 9.10.

*2-((E)-2-(Dimethylamino)vinyl)-6-((E)-4-methoxystyryl)-4H-pyran-4-one* (**2g**). The reaction was carried out for 3 h. Yield 185.5 mg (51%), brown crystals, mp 155–157 °C. IR (ATR) ν 3070, 2967, 1637, 1612, 1549, 1383, 1346, 1096, 1020, 937. ^1^H NMR (400 MHz, DMSO-*d*_6_) δ 2.96 (6H, s, 2NMe_2_), 3.80 (3H, s, OMe), 4.88 (1H, d, *J* = 13.2 Hz, -CH=C-N), 5.67 (1H, d, *J* = 2.1 Hz, CH), 6.00 (1H, d, *J* = 2.1 Hz, CH), 6.85 (1H, d, *J* = 16.2 Hz, -CH=C–Ar), 6.99 (2H, d, *J* = 8.8 Hz, H-3, H-5, Ar), 7.43 (1H, d, *J* = 16.2 Hz, =CH–Ar), 7.51 (1H, d, *J* = 13.2 Hz, =CH–N), 7.65 (2H, d, *J* = 8.8 Hz, H-2, H-6, Ar). ^13^C NMR (101 MHz, DMSO-*d*_6_) *δ* 55.2, 86.5, 103.9, 112.2, 114.3, 118.2, 128.1, 129.0, 133.4, 146.3, 159.5, 160.1, 165.4, 177.9 (the NMe_2_ group was not observed). HRMS (ESI) m/z [M + H]^+^. Calculated for C_18_H_20_NO_3_: 298.1443. Found: 298.1450.

### 3.3. Synthesis of Compounds **4**

*(E)-2-(2-(Dimethylamino)vinyl)-6-methyl-4H-pyran-4-one* (**4a**). 2,6-Dimethyl-4-pyrone (**3a**) (0.202 g, 1.63 mmol), DMF-DMA (0.583 g, 4.89 mmol) and *N*-methylimidazole (66.9 mg, 0.815 mmol) were heated at 120 °C in an autoclave for 15 h. The reaction mixture was treated with boiling *n*-heptane (40 mL) to extract product **4a**. The residue was diluted with Et_2_O to give 0.0876 g (23%) of brown needles of pyrone **5a** (mp 203–204 °C). The solution of *n*-heptane was evaporated to 2 mL, and the precipitate was filtered. Yield 0.0479 g (17%), yellow powder, mp 109–111 °C. IR (ATR) ν 3058, 2912, 1652. 1557, 1394, 1376, 1098. 913. ^1^H NMR (400 MHz, CDCl_3_) *δ* 2.21 (3H, s, Me), 2.91 (6H, s, NMe_2_), 4.71 (1H, d, *J* = 13.2 Hz, =CH(α)), 5.75 (1H, d, *J* = 2.2 Hz, CH), 5.91 (1H, d, *J* = 2.2 Hz, CH), 7.12 (1H, d, *J* = 13.2 Hz, =CH(β)). ^13^C NMR (151 MHz, CDCl_3_) *δ* 19.5, 86.8, 103.5, 112.5, 146.3, 163.2, 166.4, 178.5 (NMe_2_ was not observed). The NMR spectra are in accordance with the literature data [13].

*(E)-3-Bromo-2-(2-(dimethylamino)vinyl)-6-methyl-4H-pyran-4-one* (**4b**). 2,6-Dimethyl-3-bromo-4-pyrone (**3b**) (100 mg, 0.493 mmol), DMF-DMA (70.6 mg, 0.592 mmol), and *N*-methylimidazole (60.8 mg, 0.740 mmol) were heated in an autoclave for 8 h at 120 °C. The reaction was monitored by TLC. After completion of the reaction (TLC monitoring), the product was filtered and washed with Et_2_O (2 mL). The compound was further purified by column chromatography (CHCl_3_:EtOH = 10:0.5). Yield 102 mg (80%), yellow powder, mp 162–163 °C. IR (ATR) ν 3063, 2918, 1661, 1558, 1421, 1270, 1007, 945, 773. ^1^H NMR (500 MHz, CDCl_3_) δ 2.23 (3H, s, Me), 3.01 (6H, s, NMe_2_), 5.27 (1H, d, *J* = 13.0 Hz, =CH(α)), 6.00 (1H, s, H-5), 7.24 (1H, d, *J* = 13.0 Hz, =CH(β)). ^13^C NMR (126 MHz, CDCl_3_) *δ* 19.3, 38.3, 44.0, 87.1, 103.1, 110.7, 147.2, 161.7, 162.8, 173.6. HRMS (ESI) m/z [M + H]^+^. Calculated for C_15_H_16_NO_2_: 258.0085. Found: 258.0130.

### 3.4. General Method for the Synthesis of Bis(enamino)-substituted 4-Pyrones **5a,b**

A mixture of 2,6-dimethyl-4-pyrone **3a** or **3b** (0.806 mmol), DMF-DMA (480 mg, 4.03 mmol), and *N*-methylimidazole (33.0 mg, 0.403 mmol) was heated in an autoclave for 15 h (for **5a**) or 10 h (for **5b**) at 130 °C. Then the reaction mixture was diluted with Et_2_O (5 mL) and the product filtered.

*2,6-Bis((E)-2-(dimethylamino)vinyl)-4H-pyran-4-one* (**5a**). Yield 96.3 mg (51%), brown crystals, mp 203–204 °C. IR (ATR) ν 2990, 2810, 1640, 1615, 1542, 1360, 1335, 1095, 947. ^1^H NMR (400 MHz, DMSO-*d*_6_) *δ* 2.89 (12H, s, 2NMe_2_), 4.75 (2H, d, *J =* 13.3 Hz, =CH(α)), 5.43 (2H, s, H-3, H-5), 7.29 (2H, d, *J* = 13.3 Hz, =CH(β)). ^13^C NMR (100 MHz, DMSO-*d*_6_) *δ* 40.0–41.0 (br.s), 87.2, 103.2, 144,9, 163.5, 177.9. HRMS (ESI) m/z [M + H]^+^. Calculated for C_13_H_19_N_2_O_2_: 235.1447. Found: 235.1448.

*3-Bromo-2,6-bis((E)-2-(dimethylamino)vinyl)-4H-pyran-4-one* (**5b**). The product was additionally purified by column chromatography (CHCl_3_:EtOH = 10:0.5). Yield 134 mg (53%), dark orange powder, mp 185–186 °C. IR (ATR) ν 3398, 3020, 2916, 1630, 1487, 1353, 1105, 947, 753. ^1^H NMR (500 MHz, CDCl_3_) δ 2.91 (6H, s, NMe_2_), 2.97 (6H, s, NMe_2_), 4.79 (1H, d, *J* = 13.2 Hz, CH(α)), 5.24 (1H, d, *J* = 13.0 Hz, =CH(α)), 5.75 (1H, s, H-5), 7.01 (1H, d, *J* = 13.2 Hz, CH(β)), 7.14 (1H, d, *J* = 13.0 Hz, =CH(β)). ^13^C NMR (126 MHz, CDCl_3_) δ 40.7 (br.s, NMe_2_), 87.6, 87.9, 102.1, 102.9, 144.7, 146.2, 160.5, 163.0, 173.6. HRMS (ESI) m/z [M + H]^+^. Calculated for C_15_H_16_NO_2_: 313.0507. Found: 313.0550.

### 3.5. General Method for the Preparation of 2-((E)-4-Methoxystyryl)-6-((E)-2-aminovinyl)-4H-pyran-4-one **6a–c**

Enamino-substituted pyrone **2g** (75.7 mg, 0.255 mmol) and N-nucleophile (0.31 mmol) or *p*-phenylenediamine (13.7 mg, 0.127 mmol) were stirred at 90 °C for 3 h in AcOH (1 mL). For **6a,c**, the precipitate was filtered and washed with EtOH. For **6b**, the reaction mixture was diluted with H_2_O (5 mL). The solid that formed was recrystallized from EtOH–toluene.

*2-((E)-4-Methoxystyryl)-6-((E)-2-(phenylamino)vinyl)-4H-pyran-4-one* (**6a**). Yield 75.6 mg (86%), yellow powder, mp 229–231 °C. IR (ATR) ν 3272, 3078, 2958, 1698, 2637, 1596, 1495, 1278, 1269. ^1^H NMR (400 MHz, DMSO-*d*_6_) *δ* 3.81 (3H, s, Me), 5.57 (1H, d, *J =* 13.3 Hz, =CH(α)), 5.99 (1H, d, *J =* 1.9 Hz, CH), 6.14 (1H, d, *J =* 1.9 Hz, CH), 6.92 (1H, d, *J =* 16.3 Hz, =CH(α)ʹ), 6.93 (1H, t, *J =* 7.7 Hz, H-4 Ph), 7.01 (2H, d, *J =* 7.9 Hz, H-3, H-5 Ar), 7.16 (2H, d, *J =* 7.9 Hz, H-2, H-6 Ph), 7.31 (2H, t, *J =* 7.8 Hz, H-3, H-5 Ph), 7.50 (1H, d, *J* = 16.3 Hz, =CH(β)ʹ), 7.66 (2H, d, *J =* 8.6 Hz, H-2, H-6 Ar), 7.92 (1H, t, *J* = 12.7 Hz, =CH(β)), 9.58 (1H, d, *J* = 12.2 Hz, NH). ^13^C NMR (100 MHz, DMSO-*d*_6_) *δ* 55.7, 94.8, 107.0, 112.6, 114.9, 115.3, 118.5, 121.5, 128.5, 129.6, 129.9, 134.6, 136.3, 142.0, 160.8, 160.7, 164.4, 178.7. HRMS (ESI) m/z [M + H]^+^. Calculated for C_22_H_20_NO_3_: 346.1456. Found: 346.3980.

*2-((E)-2-(Diphenylamino)vinyl)-6-((E)-4-methoxystyryl)-4H-pyran-4-one* (**6b**). Yield 88.1 mg (82%), yellow powder, mp 225–226 °C. IR (ATR) ν 3046, 3004, 2836, 1644, 1576, 1489, 1237, 1173. ^1^H NMR (500 MHz, DMSO-*d_6_*) *δ* 3.78 (3H, s, Me), 5.07 (1H, d, *J =* 13.5 Hz, =CH(α)), 5.92 (1H, s, CH), 6.14 (1H, s, CH), 6.92 (1H, d, *J =* 16.0 Hz, =CH(α)ʹ), 7.00 (2H, d, *J =* 7.9 Hz, H-3, H-5 Ar), 7.20 (4H, d, *J =* 7.5 Hz, H-2, H-6 Ph), 7.29 (2H, t, *J =* 7.1 Hz, H-4 Ph), 7.43 (1H, d, *J* = 16.0 Hz, =CH(β)ʹ), 7.48 (4H, t, *J =* 7.2 Hz, H-3, H-5 Ph), 7.60 (2H, d, *J =* 8.6, H-2, H-6 Ar), 8.09 (1H, d, *J* = 13.5 Hz, =CH(β)). ^13^C NMR (125 MHz, DMSO-*d*_6_) *δ* 55.3, 97.0, 107.7, 112.0, 114.3, 117.9, 123.8, 125.6, 127.9, 129.0, 129.9, 134.3, 139.5, 145.5, 160.2, 160.3, 163.2, 178.2. HRMS (ESI) m/z [M + H]^+^. Calculated for C_28_H_24_NO_3_: 422.1756. Found: 422.1747.

*6,6′-((1E,1′E)-(1,4-Phenylenebis(azanediyl))bis(ethene-2,1-diyl))bis(2-((E)-4-methoxystyryl)-4H-pyran-4-one)* (**6c**). Yield 63.5 mg (82%), burgundy powder, mp 204–205 °C. IR (ATR) ν 3040, 2934, 2839, 1637, 1504, 1396, 1253, 1152. ^1^H NMR (500 MHz, DMSO-*d*_6_) *δ* 3.78 (6H, s, 2Me), 5.52 (2H, d, *J =* 13.1 Hz, =CH(α)), 5.95 (2H, d, *J =* 2.1 Hz, CH), 6.12 (2H, d, *J =* 2.1 Hz, CH), 6.92 (2H, d, *J =* 16.2 Hz, =CH(α)’), 6.97 (4H, d, *J =* 8.6 Hz, H-3, H-5 Ar), 7.15 (4H, s, Ar), 7.49 (2H, d, *J* = 16.2 Hz, =CH(β)’), 7.66 (4H, d, *J =* 8.6 Hz, H-2, H-6 Ar), 7.87 (2H, t, *J* = 12.0 Hz, =CH(β)), 9.52 (2H, d, *J* = 12.0 Hz, NH). ^13^C NMR (100 MHz, DMSO-*d*_6_) *δ* 55.7, 106.4, 112.6, 114.8, 116.9, 118.6, 128.5, 129.6, 136.3, 160.7, 164.7, 172.4, 178.6. HRMS (ESI) m/z [M + H]^+^. Calculated for C_38_H_33_N_2_O_6_: 613.2339. Found: 613.6820.

*2-((E)-2-(Benzylamino)vinyl)-6-((E)-4-methoxystyryl)-4H-pyran-4-one* (**6d**). Enamino-substituted pyrone **2g** (75.7 mg, 0.255 mmol) and benzylamine (32.8 mg, 0.306 mmol) were refluxed for 7 h in MeCN (1 mL). The reaction mixture was diluted with H_2_O (5 mL). The solid that formed was filtered and recrystallized from hexane–toluene. Yield 69.7 mg (76%), yellow powder, mp 144–145 °C. IR (ATR) ν 2900, 1651, 1520, 1394, 1250, 1169, 1027. ^1^H NMR (400 MHz, DMSO-*d*_6_) *δ* 3.80 (3H, s, Me), 4.33 (2H, d, *J =* 5.3 Hz, CH_2_), 5.04 (1H, d, *J =* 13.4 Hz, =CH(α)), 5.66 (1H, s, *J =* 2.2 Hz, CH), 6.00 (1H, s, *J =* 2.2 Hz, CH), 6.85 (2H, d, *J =* 16.2 Hz, =CH(α)ʹ), 6.99 (2H, d, *J =* 8.7 Hz, H-3, H-5 Ar), 7.25–7.31 (1H, m, Ph), 7.31–7.40 (6H, m), 7.43 (1H, br.s, NH), 7.61 (2H, d, *J =* 8.7 Hz, H-2, H-6 Ar). ^13^C NMR (101 MHz, DMSO-*d*_6_) *δ* 31.1, 55.7, 87.5, 104.9, 112.7, 114.8, 118.8, 127.6, 127.8, 128.5, 128.9, 129.5, 133.7, 159.9, 160.6, 165.8, 178.5. HRMS (ESI) m/z [M + H]^+^. Calculated for C_22_H_20_NO_3_: 346.1456. Found: 346.3980.

*2,6-Bis((E)-2-(phenylamino)vinyl)-4H-pyran-4-one* (**6e**). Pyrone **5a** (100 mg, 0.427 mmol) and aniline (99.4 g, 1.07 mmol) were stirred at room temperature for 24 h in AcOH (1.5 mL). The precipitate was filtered and washed with EtOH. Yield 77.5 mg (55%), orange powder, mp 217–218 °C. IR (ATR) ν 3221, 3054, 1657, 1645, 1544, 1493, 1278, 1148, 945. ^1^H NMR (500 MHz, DMSO-*d*_6_) *δ* 5.52 (2H, d, *J =* 13.3 Hz, =CH(α)), 5.77 (2H, s, CH), 6.92 (2H, t, *J =* 7.3 Hz, H-2, H-6 Ph), 7.13 (4H, d, *J =* 7.9 Hz, H-3, H-5 Ph), 7.87 (2H, t, *J* = 12.8 Hz, =CH(β)), 9.49 (2H, d, *J* = 12.3 Hz, NH). ^13^C NMR (126 MHz, DMSO-*d*_6_) *δ* 94.4, 106.0, 114.6, 120.8, 129.4, 135.2, 141.9, 162.5, 178.2. Anal. Calculated for C_21_H_18_N_2_O_2_: C 76.34; H 5.49; N 8.48. Found: C 76.42; H 5.56; N 8.43.

### 3.6. Synthesis of Compounds **7a,b**

*2-((E)-4-Methoxystyryl)-6-((E)-2-(2-methyl-1H-indol-3-yl)vinyl)-4H-pyran-4-one* (**7a**). Styryl-4-pyrone **2g** (105 mg, 0.353 mmol) and 2-methylindole (55.6 mg, 0.424 mmol) were refluxed in AcOH (1 mL) for 7 h. The precipitate formed was filtered and washed with EtOH. Yield 0.0835 g (62%), yellow powder, mp 284–285 °C. IR (ATR) ν 3131, 3035, 2958, 2928, 2834, 1604, 1556, 1394. ^1^H NMR (400 MHz, DMSO-*d*_6_) *δ* 2.63 (3H, s, Me), 3.82 (3H, s, OMe), 6.25 (1H, d, *J* = 2.2 Hz, CH), 6.37 (1H, d, *J* = 2.2 Hz, CH), 6.85 (1H, d, *J* = 16.2 Hz, =CH(α)), 7.00 (1H, d, *J* = 16.2 Hz, =CH(α)), 7.03 (2H, d, *J* = 8.9 Hz, H-3, H-5, Ar), 7.12–7.19 (2H, m, H-5, H-6 Ind), 7.35–7.42 (1H, m, H-7 Ind), 7.58 (1H, d, *J* = 16.1 Hz, =CH(β)), 7.68 (2H, d, *J* = 8.7 Hz, H-2, H-6 Ar), 7.79 (1H, d, *J* = 16.2 Hz, =CH(β)), 7.98–8.04 (1H, m, H-4 Ind), 11.65 (1H, s, NH). ^13^C NMR (151 MHz, DMSO-*d*_6_) *δ* 12.2, 55.8, 109.4, 111.0, 111.8, 113.1, 113.2, 115.0, 118.5, 120.1, 121.0, 122.2, 126.2, 128.4, 129.7, 135.2, 136.5, 141.4, 161.0, 161.3, 163.3, 179.3. HRMS (ESI) m/z [M + H]^+^. Calculated for C_17_H_18_NO_2_: 268.1338. Found: 268.1342.

*2,6-Bis((E)-2-(2-methyl-1H-indol-3-yl)vinyl)-4H-pyran-4-one* (**7b**). Bis(enamino)-substituted 4-pyrone **5a** (100 mg, 0.353 mmol) and 2-methylindole (55.6 mg, 0.424 mmol) was refluxed in AcOH (1 mL) for 10 h. The precipitate formed was filtered and washed with EtOH. Yield 74.6 mg (52%), orange powder, mp >270 °C. IR (ATR) ν 3178, 3056, 1631, 1611, 1539, 1399, 1274, 1153, 936. ^1^H NMR (500 MHz, DMSO-*d*_6_) *δ* 2.64 (6H, s, 2Me), 6.28 (2H, s, H-3, H-5), 6.87 (2H, d, *J* = 16.1 Hz, =CH(α)), 7.13–7.19 (4H, m, H-5, H-6 Ind), 7.36–7.41 (2H, m, H-7 Ind), 7.82 (2H, d, *J* = 16.1 Hz, =CH(β)), 7.98–8.04 (2H, m, H-4 Ind), 11.7 (2H, s, NH). ^13^C NMR (126 MHz, DMSO-*d*_6_) *δ* 11.6, 108.8, 110.7, 111.3, 112.8, 119.6, 120.5, 121.7, 125.6, 128.4, 136.0, 140.7, 162.1, 179.0. HRMS (ESI) m/z [M + H]^+^. Calculated for C_25_H_22_NO_3_: 384.1600. Found: 384.1608.

### 3.7. Synthesis of 6-(3-Phenylisoxazol-4-yl)-4H-pyran-4-ones **8a,b**

2-((Dimethylamino)vinyl)-6-styryl-4*H*-pyran-4-one **2f,g** (0.32 mmol) and *N*-hydroxybenzimidoyl chloride (60.7 mg, 0.390 mmol) were stirred for 4 days in dry 1,4-dioxane at room temperature. The precipitate that formed was filtered and washed with toluene.

*(E)-2-(4-(Dimethylamino)styryl)-6-(3-phenylisoxazol-4-yl)-4H-pyran-4-one* (**8a**). Yield 81.2 mg (66%), brown crystals, mp 199–200 °C. IR (ATR) ν 3078, 2909, 1587, 1523, 1350, 1127, 940. ^1^H NMR (400 MHz, DMSO-*d*_6_) *δ* 2.97 (6H, s, NMe_2_), 6.20 (1H, s, CH), 6.33–6.48 (2H, m), 6.65 (1H, d, *J =* 16.5 Hz, =CH(β)), 6.70 (2H, d, *J =* 7.6 Hz, Ar), 7.24 (2H, d, *J =* 7.6, Ar), 7.47–7.71 (5H, m, Ph), 9.82 (1H, s, H-5 Isox). ^13^C NMR (100 MHz, DMSO-*d*_6_) *δ* 112.3, 113.4, 113.7, 114.0, 122.7, 128.2, 129.2, 129.4, 129.5, 130.8, 136.3, 151.7, 154.7, 159.9, 162.2, 162.9, 178.5 (NMe_2_ + 1C were not observed). HRMS (ESI) m/z [M + H]^+^. Calculated for C_24_H_21_N_2_O_3_: 388.1522. Found: 388.1524.

*((E)-2-(4-Methoxystyryl)-6-(3-phenylisoxazol-4-yl)-4H-pyran-4-one* (**8b**). Yield 95.1 mg (80%), light yellow crystals, mp 221–222 °C. IR (ATR) ν 3048, 2837, 1657, 1626, 1512, 1379, 823. ^1^H NMR (400 MHz, DMSO-*d*_6_) *δ* 3.80 (3H, s, OMe), 6.27 (1H, s, CH), 6.42 (1H, s, CH), 6.46 (1H, d, *J =* 15.3 Hz, =CH(α)), 6.83 (1H, d, *J =* 15.3 Hz, =CH(β)), 6.97 (2H, br.s, Ar), 7.37 (2H, br.s, Ar), 7.45–7.78 (5H, m, Ph), 9.82 (1H, s, H-5 Isox). ^13^C NMR (126 MHz, DMSO-*d*_6_) *δ* 54.9, 112.5, 113.0, 113.3, 114.5, 127.3, 128.2, 128.5, 128.9, 129.7, 130.3, 154.8, 157.9, 169.1, 161.1, 166.4, 177.4 (2C were not observed). Anal. Calculated for C_27_H_17_NO_4_: C 74.38; H 4.61; N 3.77. Found: C 74.15; H 4.56; N 3.69.

### 3.8. Synthesis of compound **9**

*2,6-Bis(3-phenylisoxazol-4-yl)-4H-pyran-4-one* (**9**). Bis(enamino)-substituted pyrone **5a** (100 mg, 0.427 mmol) and *N*-hydroxybenzimidoyl chloride (146 mg, 0.938 mmol) were refluxed in dry dioxane (2 mL) for 4 h. The precipitate formed was filtered and washed with EtOH. Yield 64.0 mg (39%), beige powder, mp 238–239 °C. IR (ATR) ν 3066, 2890, 1657, 1612, 1549, 1445, 1404, 1143, 913. ^1^H NMR (500 MHz, DMSO-*d*_6_) *δ* 6.19 (2H, s, H-3, H-5), 7.51–7.55 (m, 8H, Ph); 7.55–7.60 (m, 2H, Ph), 9.13 (s, 2H, Isox). ^13^C NMR (126 MHz, DMSO-*d*_6_) *δ* 112.5, 113.1, 127.1, 128.6, 128.9, 130.3, 155.1, 159.1, 161.3, 177.0. Anal. Calculated for C_23_H_14_N_2_O_4_: C 72.25; H 3.69; N 7.33. Found: C 72.13; H 3.84; N 7.45.

## Data Availability

Data is contained within the article and Appendix A.

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
