# Peer review of "2-(2-(Dimethylamino)vinyl)-4H-pyran-4-ones as Novel and Convenient Building-Blocks for the Synthesis of Conjugated 4-Pyrone Derivatives"

_molecules, 2022, doi:10.3390/molecules27248996_

Round 1
Reviewer 1 Report
The research presented by Obydennov et al. describes the use of DMF-DMA as a convenient reagent for the preparation of enamines, this case, as substituents of different pyranones. In this particular case, this method is an excellent alternative for the research previously reported by Stanovnik (2008 heterocycles 899, DOI: 10.3987/COM-07-11252) in which a similar product is obtained with DMF-DMA (I recomend to cite this paper). Similarities of both research papers are limmited to a single entry.
Regarding the research presented herein, it can be applied to different pyranones with low to good yield (8-87%) and additionaly , the obtained substrates (especially 2f, 2g and 5a) are used for performing chemical manipulations with C- and N-centered nucleophiles and with dipoles (nitrile oxides). Finally, several optical properties are mesasured and explained with FMOs analysis.
The experimental part is described adequaltelly and the purity of the compounds by NMR analysis is acceptable in most of the cases.
In opninion of this referee, observing the novelty of the research and the results obtained, and having revised the experimental part, I can consider this paper publishable in Molecules. I detail some suggestions in order to be more interesting for readers: (1) Include the missed reference; (2) propose a mechanism that justifies the reaction outcome; (3) use an alcohol as NuH in scheme 5 in order to confirm the formation of vinyl ethers; (4) move some resuls of the part 2.2 Photophysical properties of products to the experimental section (maintain only the most important observations)
For all these comments I recommend to publish it without further modifications.
Author Response
We are grateful to the reviewer for a careful study of our work, as well as useful suggestions and valuable comments.
In this particular case, this method is an excellent alternative for the research previously reported by Stanovnik (2008 heterocycles 899, DOI: 10.3987/COM-07-11252) in which a similar product is obtained with DMF-DMA (I recomend to cite this paper). Similarities of both research papers are limmited to a single entry.
(1) Include the missed reference;
The reference on the Stanovnik’s review has been added (Reference 33).
(2) propose a mechanism that justifies the reaction outcome;
The yield of the enamination reaction is largely affected by the nature of the substituents in the pyrone ring, which determines the CH acidity of the initial pyrones and the possibility of side reactions.
(3) use an alcohol as NuH in scheme 5 in order to confirm the formation of vinyl ethers;
The reaction does not proceed with alcohols due to their low nucleophilicity.
(4) move some resuls of the part
2.2 Photophysical properties of products to the experimental section (maintain only the most important observations).
The results in this work are discussed rather briefly. When they are reduced, the logical connection may be broken.

Reviewer 2 Report
In the manuscript, the authors synthesized several pyrone derivatives. The synthesis is well-known and the photophysical properties studies were incomplete. The optical properties and theoretical aspects of interesting new structures are barely discussed. Thus, I would NOT recommend this manuscript to be published in Molecules. However, this paper could be published somewhere like Chemistry after addressing the concerns and comments outlined below.
1. The author has spent a lot of time describing the synthetic methodology, however, this synthesis is not novel, which is previously reported (J. Chem. Soc., Perkin Trans. 1, 1999, 3005-3013), in this contribution, the only difference is a different base.
2. For all these compounds discussed in the paper, the fluorescence in protic solvents can be related to the H-bond between the carbonyl oxygen and solvent, instead of intramolecular charge transfer.
The hydrogen bond effect between the solvent and solute should be taken into primary consideration. It’s clear that the degree of red-shifted absorption of 4a in these protic solvents is accord with the hydrogen donor ability of these protic solvents: MeOH > EtOH > iPrOH.
3. Quantum chemical calculations in these studies cannot provide meaningful explanations and information by using the CPCM implicit solvent mode. The author should place the alcohol molecules to the hydrogen bond sites and roughly account for the solute-solvent interactions. Or one may do molecular dynamic simulation.
There are no DFT studies for the new compounds like 6 and 7.
The CAM-B3LYP is not suitable for the TD-DFT these types of molecules (Unless you have considerable charge-transfer systems) since it has a very high HF component, which could overestimate the vertical excitation energies. To confirm the charge transfer, one could obtain the EDD map (for the S0 and S1 states).
4. I am surprised that the author did not conduct photoisomerization studies for these compounds.
Author Response
The synthesis is well-known and the photophysical properties studies were incomplete. The optical properties and theoretical aspects of interesting new structures are barely discussed. Thus, I would NOT recommend this manuscript to be published in Molecules. However, this paper could be published somewhere like Chemistry after addressing the concerns and comments outlined below.
We are grateful to the reviewer for a careful study of our work, as well as useful suggestions and important comments, but at the same time we do not agree with the overall assessment of the work. The results obtained are new and interesting both from the point of view of the synthetic procedure and further application of the obtained merocyanines.
- The author has spent a lot of time describing the synthetic methodology, however, this synthesis is not novel, which is previously reported (J. Chem. Soc., Perkin Trans. 1, 1999, 3005-3013), in this contribution, the only difference is a different base.
We have added the reference provided. The reference is devoted not to 4-pyrones, but to 2-methyl-chromone, which was studied in [4+2]-cycloaddition reactions. An attempt to carry out the reaction using pyridine failed to isolate the desired product.
The sentence has been added.
“The enamination with the use of pyridine as a solvent and DMF-DMA (1.2 equiv.) at 100 °C or 120 °C did not give the desired product.”
- For all these compounds discussed in the paper, the fluorescence in protic solvents can be related to the H-bond between the carbonyl oxygen and solvent, instead of intramolecular charge transfer.
The hydrogen bond effect between the solvent and solute should be taken into primary consideration. It’s clear that the degree of red-shifted absorption of 4a in these protic solvents is accord with the hydrogen donor ability of these protic solvents: MeOH > EtOH > iPrOH.
Additionally to the frontier orbitals, the EDD maps for the most intensive transitions have been added in the supporting materials to confirm the ICT effect.
Proton transfer usually results in much larger Stokes shifts (J. Am. Chem. Soc. 2021, 143, 32, 12715–12724). At the same time, we found complete suppression of the fluorescence in acetic acid.
- Quantum chemical calculations in these studies cannot provide meaningful explanations and information by using the CPCM implicit solvent mode. The author should place the alcohol molecules to the hydrogen bond sites and roughly account for the solute-solvent interactions. Or one may do molecular dynamic simulation.
In this article, the difference in the photophysical properties of mono- or bis(enamino) derivatives are primarily considered with the use of calculations. The attempt was also made to evaluate the interaction with the solvent in terms of the charge on the carbonyl group. Further in-depth study of the interaction of 4-pyrones with a solvent can be considered as a separate study.
There are no DFT studies for the new compounds like 6 and 7.
Compounds 4a and 5a have been checked as representative examples to demonstrate general trends for mono- and bisenamino derivatives.
The CAM-B3LYP is not suitable for the TD-DFT these types of molecules (Unless you have considerable charge-transfer systems) since it has a very high HF component, which could overestimate the vertical excitation energies. To confirm the charge transfer, one could obtain the EDD map (for the S0 and S1 states).
Choice of the most correct functional is often not an easy question, particularly for merocyanine dyes. Since we proposed a charge-transfer takes place in our case (and EDD maps seems to prove this), we decided to use the Cam-B3LYP range-separated functional, as it incorporates a growing fraction of exact exchange with increasing inter-electronic distance (from 19% HF exchange at short range to 65% at long range) and is believed to allow the charge-transfer phenomena to be modelled accurately (Ref. 44). Besides, the calculation with the use of the CAM-B3LYP functional is in good agreement with the experimental spectral data.
The reference [45] regarding the selection of suitable functional for merocyanines has been added.
The EDD maps for the most intensive transitions have been added in the supporting materials to confirm the ICT effect.
The sentence has been added.
“Besides, the ICT phenomena for the pyrones were confirmed with the use of EDD maps (see SM).”
- I am surprised that the author did not conduct photoisomerization studies for these compounds.
The obtained enamines are always present in a more stable E-configuration, and we did not observe the possible formation of other isomers.
The sentence about the configuration has been added.
“The 1Н NMR spectra of the obtained dimethylamino-substituted pyrones 2,4,5 demonstrate a characteristic set of two doublets of the enamino group with 3J coupling of 12.6–13.3 Hz, which indicates the E-configuration of the double bond and its partially double order due to the strong push-pull nature [31,36].”

Reviewer 3 Report
The manuscript of Obydennov et al. describes the synthesis and photophysical properties of a series of novel 2,6-disubstituted 4-pyrone derivatives and therefore, it is in the topic of Molecules. My reviewer comments concern only the synthesis and characterization of the products as I am not familiar with quantum-chemical calculations.
The manuscript is written in a very good way and the products are fully characterized. The plagiarism check shows that the results are not published before. I have only few minor notes.
The last sentence in the paragraph before Scheme 4 (page 5) “The structure of the product was confirmed by the comparison of the chemical shift of the methyl group with the starting 2,6-dimethyl-3-bromo-4-pyrone (3b) and product 4b.” is confusing. It is quite strong to say that such comparison leads to structural confirmation. The data are in congruence with the drawn structures, which are explicitly assigned by reliable methods. Better paraphrase this sentence.
It is written that in the absorption spectra of derivative 5a “two maxima are observed” but Figure 3a shows much more complicated picture. This point is relevant to be discussed. Moreover, these spectra are quite different than those of unsymmetrical derivative 4a.
The product name 6d on Scheme 5 is duplicated. Rename the second one (55%, AcOH, rt) to 6e.
It is written on Scheme 4 for derivatives 4a and 5a that “4a:5a = 17:23”, which does not give information about the reaction yield. It is clear from the experimental part that 5a and 4a are isolated in 23% and 17%, respectively. Better insert the yields bellow the structures as everywhere else.
It is difficult to see a yellow spectrum on a figure. I would suggest to replace with a more visible colour.
It is written in the general part of the experimental section that “The chemical shifts (δ) are reported in ppm relative to the internal standard … C6F6 (19F NMR),..” but the 19F spectrum of 2c is not presented.
There are some discrepancies in the data presented in different parts of the manuscript. Please, unify the following data:
The yield of derivative 2g is 51% on Table 2 and 50% in the Materials and Methods chapter.
The yield of derivative 5a is 51% on Scheme 4 and 52% in the Materials and Methods chapter.
Indicate “at rt” in the general procedure for the synthesis of derivatives 8a and 8b as done in the text.
In summary, I suggest publication of the manuscript after minor revision.
Author Response
We are grateful to the reviewer for a careful study of our work, as well as useful suggestions and valuable comments.
The manuscript of Obydennov et al. describes the synthesis and photophysical properties of a series of novel 2,6-disubstituted 4-pyrone derivatives and therefore, it is in the topic of Molecules. My reviewer comments concern only the synthesis and characterization of the products as I am not familiar with quantum-chemical calculations.
The manuscript is written in a very good way and the products are fully characterized. The plagiarism check shows that the results are not published before. I have only few minor notes.
The last sentence in the paragraph before Scheme 4 (page 5) “The structure of the product was confirmed by the comparison of the chemical shift of the methyl group with the starting 2,6-dimethyl-3-bromo-4-pyrone (3b) and product 4b.” is confusing. It is quite strong to say that such comparison leads to structural confirmation. The data are in congruence with the drawn structures, which are explicitly assigned by reliable methods. Better paraphrase this sentence.
The sentence has been paraphrased.
“The structure of the product was assigned on the basis of the chemical shift of the methyl group in comparison with the starting 2,6-dimethyl-3-bromo-4-pyrone (3b) and product 4b.”
It is written that in the absorption spectra of derivative 5a “two maxima are observed” but Figure 3a shows much more complicated picture. This point is relevant to be discussed. Moreover, these spectra are quite different than those of unsymmetrical derivative 4a.
The sentence has been corrected. The difference in the spectra is discussed in the calculation section.
“The most intense and structured band is in the short wavelength region at 300–304 nm (ε = 41400–54400 M–1cm–1), and at 378–408 nm there is a second maximum with an extinction coefficient of 17400–22300 M–1cm–1 (Figure 3, Table 4).”
The product name 6d on Scheme 5 is duplicated. Rename the second one (55%, AcOH, rt) to 6e.
The product has been renamed to 6e.
It is written on Scheme 4 for derivatives 4a and 5a that “4a:5a = 17:23”, which does not give information about the reaction yield. It is clear from the experimental part that 5a and 4a are isolated in 23% and 17%, respectively. Better insert the yields bellow the structures as everywhere else.
The yields have been added in the scheme.
It is difficult to see a yellow spectrum on a figure. I would suggest to replace with a more visible colour.
The yellow color has been changed.
It is written in the general part of the experimental section that “The chemical shifts (δ) are reported in ppm relative to the internal standard … C6F6 (19F NMR),..” but the 19F spectrum of 2c is not presented.
The 19F spectrum is present in the experimental part.
There are some discrepancies in the data presented in different parts of the manuscript. Please, unify the following data:
The yield of derivative 2g is 51% on Table 2 and 50% in the Materials and Methods chapter.
The yield of derivative 5a is 51% on Scheme 4 and 52% in the Materials and Methods chapter.
Indicate “at rt” in the general procedure for the synthesis of derivatives 8a and 8b as done in the text.
Corrections have been made in accordance with the recommendations of the reviewer. The data have been verified.

Reviewer 4 Report
This article describes synthesis of 2-(2-(dimethylamino)vinyl)-4H-pyran-4-ones and their use as convenient building blocks to access novel push-pull 4-pyrone derivatives exhibiting important photophysical properties. In my opinion, this work can be published in Molecules after the following points will be addressed.
Notes/remarks for the authors:
1) In abstract and conclusion, “hetaryl-substituted” should be replaced with “isoxazolyl-substituted” since only one type of heterocycles is present
2) Page 7, “2,6-bis(oxadiazolyl)-4-pyrone” should be replaced with “2,6-bis(isoxazolyl)-4-pyrone”
3) Page 10, “located in the in the range” should be corrected
4) In calculation section, abbreviations FC and ICT should be spelled out
5) Title of section “Conclusion” is missed
6) 13C NMR spectrum for compound 6e should be added in the experimental part and SI.
7) 13C NMR spectrum for compound 2g should be added in the experimental part.
8) According to the copies of NMR spectra, brominated compounds 4b and 5b are not pure and should be repurified.
Author Response
We are grateful to the reviewer for a careful study of our work, as well as useful suggestions and valuable comments.
1) In abstract and conclusion, “hetaryl-substituted” should be replaced with “isoxazolyl-substituted” since only one type of heterocycles is present
The sentence has been corrected according to the recommendation the reviewer.
2) Page 7, “2,6-bis(oxadiazolyl)-4-pyrone” should be replaced with “2,6-bis(isoxazolyl)-4-pyrone”
The sentence has been corrected according to the recommendation the reviewer.
3) Page 10, “located in the in the range” should be corrected
The sentence has been corrected according to the recommendation the reviewer.
4) In calculation section, abbreviations FC and ICT should be spelled out
The abbreviations have been added to the text according to the recommendation the reviewer.
5) Title of section “Conclusion” is missed
The word has been added.
6) 13C NMR spectrum for compound 6e should be added in the experimental part and SI.
The 13C spectrum has been added.
“2,6-Bis((E)-2-(phenylamino)vinyl)-4H-pyran-4-one (6e). Pyrone 5a (100 mg, 0.427 mmol) and aniline (99.4 g, 1.07 mmol) were stirred at room temperature for 24 h in AcOH (1.5 mL). The precipitate was filtered and washed with EtOH. Yield 77.5 mg (55%), orange powder, mp 217–218 °C. IR (ATR) ν 3221, 3054, 1657, 1645, 1544, 1493, 1278, 1148, 945. 1H NMR (500 MHz, DMSO-d6) δ 5.52 (2H, d, J = 13.3 Hz, =CH(α)), 5.77 (2H, s, CH), 6.92 (2H, t, J = 7.3 Hz, H-2, H-6 Ph), 7.13 (4H, d, J = 7.9 Hz, Н-3, Н-5 Ph), 7.87 (2H, t, J = 12.8 Hz, =CH(β)), 9.49 (2H, d, J = 12.3 Hz, NH). 13C NMR (126 MHz, DMSO-d6) δ 94.4, 106.0, 114.6, 120.8, 129.4, 135.2, 141.9, 162.5, 178.2. Anal. Calculated for C21H18N2O2: С 76.34; Н 5.49; N 8.48. Found: С 76.42; Н 5.56; N 8.43.”
7) 13C NMR spectrum for compound 2g should be added in the experimental part.
The 13C spectrum has been added.
“2-((E)-2-(Dimethylamino)vinyl)-6-((E)-4-methoxystyryl)-4H-pyran-4-one (2g). The reaction was carried out for 3 h. Yield 185.5 mg (51%), brown crystals, mp 155–157 °C. IR (ATR) ν 3070, 2967, 1637, 1612, 1549, 1383, 1346, 1096, 1020, 937. 1H NMR (400 MHz, DMSO-d6) δ 2.96 (6H, s, 2NMe2), 3.80 (3H, s, OMe), 4.88 (1H, d, J = 13.2 Hz, -CH=C-N), 5.67 (1H, d, J = 2.1 Hz, CH), 6.00 (1H, d, J = 2.1 Hz, CH), 6.85 (1H, d, J = 16.2 Hz, -CH=C–Ar), 6.99 (2H, d, J = 8.8 Hz, H-3, H-5, Ar), 7.43 (1H, d, J = 16.2 Hz, =CH–Ar), 7.51 (1H, d, J = 13.2 Hz, =CH–N), 7.65 (2H, d, J = 8.8 Hz, H-2, H-6, Ar). 13C NMR (101 MHz, DMSO-d6) δ 55.2, 86.5, 103.9, 112.2, 114.3, 118.2, 128.1, 129.0, 133.4, 146.3, 159.5, 160.1, 165.4, 177.9 (the NMe2 group was not observed). HRMS (ESI) m/z [M + H]+. Calculated for C18H20NO3: 298.1443. Found: 298.1450.”
8) According to the copies of NMR spectra, brominated compounds 4b and 5b are not pure and should be repurified.
The products have been repurified. The spectrum of 4b has been replaced. The yield of 5b has been corrected.

Round 2
Reviewer 2 Report
Most of these explanations are reasonable and make sense.
However, in Reply 2:
'...Proton transfer usually results in much larger Stokes shifts (J. Am. Chem. Soc. 2021, 143, 32, 12715–12724)...'' Excited-state proton transfer (ESPT) is different from H-bonding with surrounding solvents. The former has formed a new structure.
Since the authors have admitted these molecules follow Kasha's rule, we only need to investigate the s0 to s1 transition EDD maps.
Author Response
Dear Editor, we are sending a revised version of our manuscript “2-(2-(Dimethylamino)vinyl)-4H-pyran-4-ones as Novel and Convenient Building-Blocks for the Synthesis of Conjugated 4-Pyrone Derivatives”.
We are grateful to the referee for a careful study of our work, as well as for valuable remarks and useful suggestions. We have corrected the text according to the recommendation of the reviewer and some of them we will definitely take into account in our future works.
Our answers in the text are colored in blue.
Sincerely, Dr. Obydennov Dmitrii L.
Reviewer 2
Most of these explanations are reasonable and make sense.
However, in Reply 2:
'...Proton transfer usually results in much larger Stokes shifts (J. Am. Chem. Soc. 2021, 143, 32, 12715–12724)...'' Excited-state proton transfer (ESPT) is different from H-bonding with surrounding solvents. The former has formed a new structure.
We agree with the reviewer about the solvation. We've corrected the text to emphasize the hydrogen bonding. The reference regarding the solvation and the charge transfer has been added (Ref. 39).
“For all excited states, an increase in the negative charge on carbonyl oxygen is observed (except for 4a in vacuum), which is in good agreement with the change in dipole moments. The strongest charge changes during the GS→S1 transition were found in MeOH (0.093 for 4a, 0.149 for 5a). This result is related to the ICT effect in this solvent, which is more pronounced for pyrone 5a, and determines the stabilization of the excited states via the hydrogen bonding interaction between the carbonyl group and alcohol molecules [39].”
Since the authors have admitted these molecules follow Kasha's rule, we only need to investigate the s0 to s1 transition EDD maps.
The EDD maps have been added for the S0→S1 transitions in SM to confirm the ICT effect.
